# The Needs of LGBTI People Regarding Health Care Structures, Prevention Measures and Diagnostic and Treatment Procedures: A Qualitative Study in a German Metropolis

**DOI:** 10.3390/ijerph16193547

**Published:** 2019-09-22

**Authors:** Ute Lampalzer, Pia Behrendt, Arne Dekker, Peer Briken, Timo O. Nieder

**Affiliations:** Institute for Sex Research, Sexual Medicine and Forensic Psychiatry, University Medical Center Hamburg-Eppendorf, 20246 Hamburg, Germany; p.behrendt@uke.de (Pi.B.); dekker@uke.de (A.D.); briken@uke.de (P.B.); t.nieder@uke.de (T.O.N.)

**Keywords:** diversity, gender, health care system, homosexuality, LGBTI

## Abstract

(1) Background: Studies indicate that lesbian, gay, bisexual, transgender and intersex (LGBTI) people constantly face challenges and disadvantages in the health care system that prevent them from getting the best possible patient-centered care. However, the present study is the first to focus on LGBTI-related health in a major German metropolis. It aimed to investigate health care structures, prevention measures and diagnostic as well as treatment procedures that LGBTI individuals need in order to receive appropriate patient-centered health care and health promotion. (2) Methods: Following a participatory approach, five expert interviews with LGBTI people with multiplier function, i.e., people who have a key role in a certain social milieu which makes them able to acquire and spread information in and about this milieu, and three focus groups with LGBTI people and/or health professionals were conducted. Qualitative data were analyzed according to the principles of content analysis. (3) Results: The specific needs of LGBTI individuals must be recognized as a matter of course in terms of depathologization, sensitization, inclusion, and awareness. Such an attitude requires both basic knowledge about LGBTI-related health issues, and specific expertise about sufficient health care services for each of the minorities in the context of sex, sexual orientation and gender identity. (4) Conclusions: For an appropriate approach to LGBTI-centered health care and health promotion, health professionals will need to adopt a better understanding of specific soft and hard skills.

## 1. Introduction

“The mission of the International Psychology Network for Lesbian, Gay, Bisexual, Transgender and Intersex Issues (IPsyNet) is to facilitate and support the contributions that the discipline of psychology makes to a global understanding of human sexuality and gender diversity so as to ensure the health and well-being of people around the world who identify, or are perceived as, lesbian, gay, bisexual, transgender, intersex, queer, or sexually and gender diverse people (LGBTIQ+)” [1]. The IPsyNet consists of psychological organizations from different countries from around the world and their Statement on LGBTIQ+ Concerns states: “LGBTIQ+ identities and expressions are normal and healthy variations of human functioning and relationships” [1]. In line with this, guidelines, policy statements, and human rights initiatives emphasized in recent years that ignorance, discrimination, stigmatization and lack of knowledge are major problems in the health care of lesbian, gay, bisexual, transgender and intersex (LGBTI) people. The term transgender is used as an umbrella term for people whose gender identity does not (completely) correspond with their sex characteristics (e.g., people who identify as trans, transsexual, nonbinary, or genderqueer). Intersex is used as an umbrella term to denote a number of different variations (chromosomal, gonadal, hormonal, and phenotypical) in a person’s innate bodily characteristics that do not all correspond with one or the same sex. LGB people can be cisgender (meaning that they feel that their gender is congruent with their sex characteristics), transgender, and intersex people. Guidelines, policy statements, and human rights initiatives aimed at tackling health disparities [2,3] to better ensure the health and well-being of LGBTI people [1,4], and called for LGBTI people’s right to health [5,6,7,8,9]. The APA “Guidelines for Psychological Practice with Lesbian, Gay, and Bisexual Clients” highlight the need to understand the effects of stigma and the recognition of the unique experiences and challenges of lesbian, gay and bisexual clients [2]. The “Agenda 2030 for LGBTI Health and Well-Being” calls for commitment “to end stigma and discrimination based on sexual orientation, gender identity and expression, and sex characteristics (…) in the provision of health care services, including prevention, promotion, and treatment” [4]. “HIV & Aids”, “Mental Health & Well-Being”, “Drug & Alcohol Use”, “Sexual & Reproductive Health”, “Universal Health Coverage”, “Access to Affordable Medicines”, and “Training of the Health Workforce” are named as important issues in LGBTI health and well-being [4]. Furthermore, a wide range of empirical research shows that these issues are specifically relevant for LGBTI individuals [9,10,11,12,13,14,15,16,17,18,19,20,21,22,23,24,25,26,27,28,29,30]. In summary, there is a call for equal opportunities for LGBTI individuals and heterosexual cisgender people in health care treatment on the one hand, as well as a need for specific patient-centered services for LGBTI individuals on the other hand.

Although many issues affect all LGBTI groups, it is nonetheless important to differentiate between gender identity and sexual orientation: “(…) gender identity and sexual orientation are distinct but interrelated constructs. (…) *Sexual orientation* is defined as a person’s sexual and/or emotional attraction to another person (…), compared with *gender identity*, which is defined by a person’s felt, inherent sense of gender” [3].

Medical guidelines for gender incongruence, gender dysphoria, and transgender health emphasize the need to take psychological, physical, social, and cultural aspects into consideration in the context of diagnostic and treatment procedures. Transition-related health care, such as mental health care (e.g., assistance to explore one’s gender identity), endocrine care (e.g., sex hormones), surgeries (e.g., breast and/or genital reconstructive surgery), and so forth, should be tailored to the individual´s needs [31,32,33]. Medical guidelines for the management of individuals with intersex conditions demand good medical care (e.g., careful clinical and biochemical evaluation, genetic counselling, longitudinal assessment). They also point out the importance of informed consent and psychosocial support, including peer support, mental health care, and communication skills training for health professionals [34,35,36,37]. Care for intersex and transgender clients requires interdisciplinary cooperation between different medical disciplines and mental health services [38,39]. Thus, medical and mental health care needs are strongly interlinked.

Several international and national qualitative studies have already investigated the health care needs of LGBTI people [9,17,18,19,20,21,22]. However, the present study is the first to investigate the health care needs of LGBTI people in Hamburg, a major German metropolitan city. Thus, it is still unknown whether the results of existing studies also prove true for health care in Hamburg. The present study is part of a larger research project about the challenges and problems with health promotion and health care for LGBTI people in Hamburg, Germany [40,41,42]. In 2017, the senate of Hamburg adopted an action plan for gender and sexual diversity, which contained eleven areas of activity for people between childhood and ninety years of age. The plan was a token of tolerance and openness in Hamburg. Financial resources were provided for 90 measures. With regards to content, the plan’s aims were information, education, sensitization, making different concepts of living more common, and the protection of rights [43]. The present study is part of this action plan.

According to the relevant specialist literature, access to care, discrimination, knowledge and dissemination of knowledge about LGBTI health, as well as awareness and sensitivity regarding LGBTI communities, are prominent issues concerning the health care needs of LGBTI people [44,45]. These issues are reflected in the topics of the expert interviews and focus groups. The present study focuses on health care structures, prevention measures, and diagnostic and treatment procedures. Therefore, the research question is: what health care structures, prevention measures, and diagnostic as well as treatment procedures do LGBTI individuals need in order to receive needs-based health care and health promotion? The common needs of LGBTI individuals as well as the specific needs of each target group (L, G, B, T, and I) will be addressed.

## 2. Materials and Methods

This study was conducted at the Institute for Sex Research, Sexual Medicine and Forensic Psychiatry at the University Medical Center Hamburg-Eppendorf as part of a gender equality policy framework program of the Free and Hanseatic City of Hamburg (FHH). It took place from April 2017 to January 2018 and was approved by the ethics committee of the Chamber of Psychotherapists Hamburg (04/2017-PTK-HH).

The present study took a qualitative approach. We were interested in specific and detailed experiences of professionals and/or clients in the health care system and aimed at discovering and understanding how the participants view their living conditions under these circumstances. Thus, a person-centered and participatory approach seemed to be appropriate to learn more about factors influencing LGBTI health in Hamburg [46,47]. The main purpose was to collect what problems and challenges LGBTI people face in Hamburg’s health care.

Data comprised five expert interviews with LGBTI people conducted in June and July 2017 and three focus groups with three to six participants in September 2017. Prior to participation, participants were provided with information about this study, including the approximate length of time for participation, data protection, and the goals of this study. Afterwards, they were asked to give informed consent. All audio recordings were transcribed and pseudonymized. Personal data that would have enabled inferences about specific interviewees were not recorded. Participants were able to withdraw from this study during and after their participation without explanation. They would have been identified by an individual code generated at the beginning of the interviews. No one withdrew from participation.

According to Denzin’s [48] basic types of triangulation, the present study ensured data triangulation via multiple data sources, i.e., experts who were interviewed because of their function as multiplier key people and people in their role as private individuals who took part in focus groups. Investigator triangulation was accomplished via integrating three researchers in the process of data collection, data analysis and interpretation. Methodological triangulation was assured when using two different qualitative methods of data collection, i.e., expert interviews and focus groups.

### 2.1. Expert Interviews

Based on current research literature, an interview guideline (Appendix A) was developed including the most prominent issues [42]:1.What kind of observable disadvantages in specific treatment/advice situations exist?

In the context of information about specific groups of the LGBTI community:2.To what extent does a lack of knowledge about each group persist?3.How available is the information on specific groups?

For the expert interviews, professional key people of the focused target groups were recruited in the Hamburg area, who in addition to their own health care experiences knew of the experiences of other people in the respective group with medical and mental health professionals (as a multiplier, i.e., as a person who is able to acquire and spread information in and about their milieu). Email requests were sent to relevant counselling centers, specialized practices, interest groups, authorities and self-help organizations. Finally, participants were recruited from BiNe e. V. (expert for bisexual people (EB)), Charlotte e. V. (expert for lesbian women (EL)), Hein and Fiete (expert for gay men (EG)), Intersexuelle Menschen e. V. (expert for intersexual people (EI)) and Magnus-Hirschfeld-Zentrum (mhc) (expert for transgender people (ET)). The interviews took place at the Institute for Sex Research or, if desired, at the workplace or home of the experts and lasted between 43 and 72 minutes. All of the interviews were conducted by the same researcher of the team (U.L.) to ensure a comparable standard. A second researcher of the team (Pi.B.) took part as a participant observer to reveal potential differences in the approach of the interviewees and decrease potential biases in the further research process.

Data were analyzed using the software tool ATLAS.ti, following the iterative process of qualitative content analysis [49]: the interviews were read several times by three researchers (U.L., Pi.B., T.O.N.). Based on the research questions, these three researchers built a system of categories by initial coding and rechecking the interview material. In the end, the results were put together thematically in order to frame the main and subcategories. Frequent comparisons and adjustments of the results by the three independent assessors ensured the interrater reliability of the results.

### 2.2. Focus Groups

Based on qualitative data of the expert interviews conducted previously, a discussion guide was developed which covered the following three main topics:How should each of the LGBTI-groups be addressed so that they can use the health system when needed?With what attitude should skilled professionals approach their clients from the LGBTI groups?To what extent does the community influence the use of health care?

The focus groups were conducted in the same way as the expert interviews (see above); representatives of the LGBTI groups and experts from professional practice were contacted. Since the focus was on personal experiences, the function as a multiplier key person was not a prerequisite for participation. Therefore, one participant was recruited by visiting a regular meeting for bisexual people and another responded via an advertisement on an internet platform. The focus groups involved three to six people aged between 30 and 63 years (focus group 1 included, by their own definition, a queer lesbian cis woman (F1L), a gay cis man (F1G), and a hetero bisexual inter man (F1B); focus group 2 included a questioning cis woman, a lesbian woman (F2L), a gay transman (F2T), and a hetero–pansexual cis inter man (F2I); focus group 3 included a heterosexual cis woman, a lesbian woman (F3L), a gay man (F3G), a bisexual woman (F3B), a gay transman (F3T) and an intersex man (F3I)). (Since the overarching project [42] also addressed heterosexual cisgender people, a questioning cis woman and a heterosexual cis woman also took part in the focus groups. However, the present evaluation refers solely to statements made by LGBTI people.)

The focus groups took place at the Institute for Sex Research and were moderated by the principal investigator (T.O.N.) and the research fellow (U.L.). The student assistant (Pi.B.) documented the process as a participating observer in order to ensure that possible biases due to the personalities of the moderators were revealed and decreased in the further process of investigation. The discussions lasted between 93 and 105 minutes.

The evaluation procedure of this data was the same as for the expert interviews. The already existing category system of the expert interviews was used for the evaluation, which consequently was again checked for its suitability, validated and extended by one subcategory (9.4). The results presented in this paper refer to the categories “requests—structures”, “availability (lacking)”, “prevention measures”, “diagnostic procedures”, and “treatment procedures”, as these are the categories that are directly linked to the research questions of this study. The category system is set out in Table 1. The direct quotes were translated by T.O.N. (Since the original quotes were in German, T.O.N. and an English proofreader looked for the best possible translation).

## 3. Results

Four issues were identified by the interviewees and will be presented in more detail: 1. health care structures, 2. human resources, 3. prevention measures, and 4. diagnostic and treatment procedures.

### 3.1. Health Care Structures

Interviewees of all LGBT groups complained about too little psychosocial and/or psychotherapeutic care offered for LGBTI issues.
**EG:** “There is a severe shortage of psychologists and psychiatrists in Hamburg [especially compared to Hamburg’s well-established network of specialized medical practices for HIV/STI, UL].”

Interviewees from all LGBTI groups demanded more contact points in the health care system specialized for the group’s specific needs.
**F2B:** “A bisexual person is not interested in gay counselling […] or a lesbian coffee shop or similar”.

The introduction of an LGBTI certificate for health care professionals who are trained in LGBTI issues was proposed as an approach for making LGBTI people’s access to adequate health care easier.
**F1G:** “[…] the largest operator of seniors’ and nursing homes in Munich just started a model project […] and presents the rainbow flag on its website […]. Unfortunately, this does not mean much, as it is still unknown how gays, lesbians, inter- and trans-people are approached. It is different in Holland. They have a certification procedure […], which is also examined by a third party.”
**F1I:** “This has to be accepted by both sides, the community and [health care professionals].”

Lesbian interviewees voiced the need for more information materials concerning lesbian health care concerns so that lesbians are sensitized accordingly and make use of health care services appropriate to their needs. F1L also expressed the need for more research in the field of lesbian health. In addition, she recommended implementing a permanent contact person who knew about LGBTI health and could be asked for advice in medical centers.
**F1L:** “If there are any reasons why they [other professionals] do not address such [LGBTI-related] issues, then I [as an LGBTI person] could do it.”

EB explained that there was a lack of visible health care services for bisexual people. He highlighted the need for low-level counseling especially.
**EB:** “A low-level counselling service, that’s it. (…) Where fears can be reduced by receiving answers to questions that are asked frequently: ‘What is happening to me? I am currently changing, do I have to be afraid? (…) So that all those who think ‘ah, I feel bad’, have a service that helps them and makes them feel recognized.”

ET also said that there was a lack of low-level health care services. He complained about long waiting times for physician and psychotherapist appointments and also mentioned a lack of couple therapy and gynecological services for transgender people. Furthermore, transgender interviewees stressed the importance of quality control, particularly because transgender people depended on referrals from mental health professionals in order to get treatment for gender dysphoria. F3T pointed to problems with the reimbursement from health insurances.
**F2T:** “You can’t chose freely as with a general physician. You are happy when you find the right professional and then you try hard (…) to make it work, even though it’s actually invasive or unprofessional or just doesn’t fit. (…) Especially in the field of transgender, where the people concerned depend on getting a referral to go on hormones.”
**F3T:** “This simply cannot be accounted for by the health system, because some treatments are linked to gender. Why should a transman go to a gynecologist? That doesn’t make any sense at all.”

EI, as well as EB, stressed the need for visible health care services, in this case for intersex health care. In this respect, she also highlighted that more peer support was needed, as recommended by the guidelines for intersex care. Moreover, she said that health care services should be evaluated and that quality management was required. Furthermore, she stated that there was no health care service for correct care of a neovagina. F1I complained about the struggle to get the complete patient records of all intersex treatment measures of the past. Similar to F3T, F3I mentioned serious problems with the payment and settlement system.
**F3I:** “Then I went to an endocrinologist and wanted to substitute testosterone, but I was supposed to pay for testosterone myself, because I was assigned female and therefore can’t get testosterone. (…) At one point, they understood and now they reimburse the costs regularly.”

### 3.2. Prevention Measures

For all the different perspectives, cancer prevention was a topic of importance. EL said that in lesbian women the “standard reasons” of heterosexual women for visiting a gynecologist (pregnancy, contraception) did not exist. However, since, for example, the same cancer screening was indicated as for heterosexual women, it was important to provide lesbian women with additional information. EL also mentioned that among lesbian women, the frequency of smoking, alcohol consumption and problem drug use was higher than among heterosexual women—a fact that also shows a need for prevention measures.

EG did not mention cancer prevention but highlighted the need for STI prevention and addiction prevention, also because there was a trend for chemsex (i.e., having sex under the influence of synthetic drugs) in the gay scene. Moreover, he said that the suicide rate was higher among gay men than among heterosexual men and talked about a self-help group called “gay and depressive”, thus pointing to the need for suicide and depression prevention.

ET pointed out that in basic health care, he did not know about any prevention programs but only knew about self-organized prevention programs from the transgender activist scene. Transgender interviewees highlighted the need for gynecological and urological prevention measures that were often overlooked with regard to transgender people. In addition, ET expressed the need for prevention of harm, e.g., for transmen who used breast binders.
**ET:** “What happens after a mastectomy with the breast cancer screening? Will it still be performed or not? (…) Urological topics—what about them? Is it clear to people that they still have to take screenings? Are they still actively invited?”
**ET:** “The whole issue of ‘breast binding or not ‘, i.e., prevention of harm. (…) Just to find a good way to deal with the own body.”

EI made a plea for adequate counselling regarding degeneration risks for parents of intersex children.
**EI:** “If I have a child who has a risk, a 32% risk of degeneration of hormone-producing organs (…), then I am shocked. But when I am told that my child has a 32% risk of becoming ill with this organ in the second half of his life from the age of 40, then this is a problem that we will have to look at later. But that’s what it’s all about. Just to give professional counselling.”

### 3.3. Diagnostic Procedures

Concerning the diagnostic procedures and issues of openness, heedfulness, physical and psychological health risks of LGBTI people, comorbidities and differential diagnostics were addressed.

F1L advocated authenticity and openness in cases of a lack of expertise, e.g., in the context of LGBTI health. She also pointed to the reality medical doctor having to deal with limited timeslots for each patient.
**F1L:** “I say ‘Okay, this is important, I have to do research, I have to inform myself. Please come back.’”
**F1L:** “I don’t think there’s time for that in a regular situation like this. I don’t know which physician also asks about the mental state. So, if anybody comes to me and suffers pain, I treat it.”

EG stressed that nicotine and alcohol consumption as well as the number of suicides were higher among gay men than among heterosexual men. He pointed out that considering not only medical-physical but also medical-psychological aspects and possibly the need for psychological care was important when seeing gay men.
**EG:** “A relationship that you had for ten years before you were out, and then ends, has a different story than a heterosexual relationship. (…) It might express itself in stomach pains and something like that. Then you go to the doctor who treats you with stomach pills. But the symptom is actually a different one—and it’s about being sensitive to it.”

F1G stated that knowing about a patient’s homosexuality might be relevant in the process of diagnostics.
**F1G:** “If there really are specific health problems, I say, ‘Okay, I’m homosexual. Please note that. This might be important to know.’”

EB mentioned that especially in the first stages of coming out, there was a higher risk of psychosomatic problems.
**EB:** “So, the most common is definitely depression—and sleep disorders. (…) And the coming out is a big topic for many: ‘I somehow decide on something. And what do I choose?’ And this confusion is great and leads to all sorts of psychosomatic symptoms.”

ET highlighted the problem that for transgender people, common standards for males and females might not be applicable and had to be taken into consideration. Moreover, he stressed that precise differential diagnostics and carefully dealing with comorbidities was highly important. In this respect, he also mentioned that there was the danger that somatic health professionals totally ignored psychological aspects.
**ET:** “In the manic phase, is this just the acting-out of trans femininity or does it simply belong in the psychotic sphere? (…) Or something like that: That belongs to trans, that belongs to eating disorders, that belongs to fear and panic, that belongs to depression. (…) And then to look at the group of diagnoses. (…) And the separation of ‘Is this acutely related to my transition’ (…)—or is it related to other issues where Trans also plays a role, and I have to interrupt some hormones somehow. (…) And with the medical doctors (…), I have the feeling that this is still completely different, because they say: ‘Well, everything that has to do with the psyche—we just need the referral letter.”

Apart from that, ET stressed that a transman’s desire to have children should not prevent experts from giving an indication for transgender treatment.

EI stressed a similar issue when expressing that carefully questioning causality was necessary, namely if psychological problems were caused by medical issues, or rather the other way around.
**EI:** “This mental side has its origin very often in a preceding medical treatment [EI for example refers to medically unnecessary surgeries on intersex children in early childhood—such as genital surgeries and gonadectomies with the effect of a need for a lifelong treatment of synthetic sex hormones –, often without a precise patient education even in adulthood].”

F1I mentioned the issue of coming out as intersex towards health professionals and made clear that for some patients it was easy whereas others were shy or ashamed. In addition, he stressed that in the context of intersex being classified as a rare disease, abusive curiosity had to be banned absolutely.
**F1I:** “What also does not work is the demonstration of affected people in the hospital: ‘Ah, you are a Klinefelter. Can we see your testicles? Yes, for a moment.’ (…) I know of cases where that was very distressing.”

### 3.4. Treatment Procedures

Regarding treatment procedures, the interviewees on the one hand named contexts in which the sexual orientation should definitely be considered and on the other hand contexts in which it should more or less be ignored. Moreover, the fear of discrimination, the wish for open-mindedness and knowledgeable treatment came up as issues of concern.

EL expressed the need for providing lesbians good support in getting fertility treatment when they asked for it. She stressed that there was a lack of respective services especially in Hamburg:
**EL:** “Well, in terms of fertility treatment, Hamburg lags behind Munich and Erlangen—and Berlin anyway.”
**EL:** “And there are countries where I assume that they are not lesbian-friendly, but where reproductive medicine is still handled openly and liberally, so there are no barriers.”

In the context of pregnancy, F1L underlined that gynecologists should not be more concerned about the family situation of lesbian women than of heterosexual women. She stated that for good health care during pregnancy, sexual orientation was not important.
**F1L:** “Well, I have two children—and then again: ‘Who is the father, how did you do that?’ Which doesn’t matter at all—nor does it matter in health care during pregnancy.”

EG stressed that psychotherapists who treated gay men should be able to openly talk about sexuality, e.g., anal intercourse, because in case of tabooing the gay man’s sexuality, effective psychotherapeutic treatment was impossible. Statements of F1G pointed to the fact that there were cases of medical treatments where sexual orientation was irrelevant, e.g., treatment of hemorrhoids at the proctologist, dental treatment, and influenza treatment, cases where it was relevant, e.g., for getting prophylaxis against hepatitis C, and cases where it might be relevant but should not be because otherwise it might be discrimination.
**F1G:** “The treatment and examination [i.e., prostate biopsy], (…) everything was no problem. I didn’t notice anything afterwards, (…) whether he was more reserved or treated me differently.”
**F1G:** “Well, in the hospital it was obvious to the staff that a man only has men visiting. But that didn’t have any negative effects.”

EB highlighted that a psychotherapist who treated a bisexual person could not hold the (wrong) view that bisexuality did not exist, a common assumption of the past that still sometimes existed. Apart from that, he stressed that bisexuality was no illness and did not require any treatment, but many bisexual people sought counseling anyways.

ET made a plea to use the whole scope of action in transgender treatment creatively and also trust the clients’ abilities to make their own decisions in a self-determined way. He also mentioned that taking the time to ‘wait and see’ could be important for the process of a transgender person’s self-development. In addition, in transgender treatment, he stressed the integration of all physical changes into the need to be cared for sufficiently, e.g., by also offering body therapy. F2T stressed that, if necessary, it was important to consider the effects of transgender treatment, but if not necessary, he did not want to explain anything about it. He mentioned that when he was at the beginning of his transition process, he was relieved that his gynecologist did not let him sit in the waiting room for long.
**F2T:** “My personality is none of his [the doctor’s] business or what my hobbies are or anything like that. But I want to be treated.”
**F2T:** “Still at the very beginning of my transition, I was at the gynecologist’s and suddenly it was my turn, otherwise I always had to wait for ages. Well, I guess this is also a situation that usually gets rather uncomfortable when you sit there for an hour in the waiting room.”

Intersex interviewees demanded that harmful treatment practices of the past, such as no ensured informed consent before treatment, had be disestablished. EI said that continuous support and practical psychosocial counselling was important in intersex care. Last but not least, she made a claim to always inform about all the treatment options, also the option of non-treatment.
**EI:** “Non-treatment as a treatment option for example (…), and accompanying it. You have to stand it. This is much more difficult than doing something quickly. (…) We don’t know what we are doing, but we are doing it. Instead of saying: ‘No, we don’t know what will happen. Your child is so individual, we don’t know that at all—and let’s wait together, and we’ll make sure together that he’s fine.’”

F2I stressed the problem of using the right reference values when treating intersex people.
**F2I:** “There are female reference values on my laboratory sheet because I have a female civil status. However, for me the male reference values are more valid because I am under testosterone. (…) And then it says: ‘But this value is too high and it is too low’. And then I had an endocrinologist, to whom I have to explain again with each treatment that this reference value should not be taken as a basis. This is tedious for me.”

## 4. Discussion

The purpose of this study was to investigate the needs of LGBTI people regarding health care structures, prevention measures, and diagnostic as well as treatment procedures. Moreover, this study aimed at analyzing needs that affect both all LGBTI people and particular groups (L, G, B, T, and I). All in all, the participants did not differ very much from each other in their positions. Differences were mainly as a result of the examples given by the interviewees.

Consistent with topics included in relevant reference books on LGBT health, the following subjects were brought up: the health-related institutional culture and climate, mental health, health risk behaviors, substance use, suicide risk, internalized homophobia and disclosure, HIV and sexually transmitted infections, cancer, urologic and gynecologic care, interdisciplinary care, and parenting [44,45]. The following topics, although regarded as relevant in specialized literature [44,45], were not explicitly mentioned: obesity, chronic illnesses, such as hypertension, asthma, and diabetes, intimate partner violence, living with disabilities, racial and ethnic minority populations, and aging. It is probably due to the interview partner’s professional and/or personal background that these topics were not the focus. Compared to previous qualitative research on the needs of LGBTI people, the results of the present study are quite comparable. The fear of discrimination, lack of knowledge of health care professionals, higher risk of mental health problems, fear of disclosure, being confronted with unquestioned heteronormative assumptions, importance of visibility, and pathologization were also highlighted as important issues in previous studies conducted in the European Union [17,18], Australia [20], and the United States [19]. By contrast, with research from Zimbabwe, problems of stigmatization, discrimination, ill-informed personnel and lacking access to health care were much less prominent and serious, although also reported in Hamburg [21,22]. Previous evidence is extended by the present study and enriched by further qualitative content, e.g., personal experiences and concrete examples, from Hamburg, Germany.

Hereinafter, the needs voiced in the present study are discussed, classified into the needs of all target groups, the needs of LGB individuals, and the needs of transgender and intersex people.

### 4.1. Needs of all Target Groups

Regarding health care structures, needs expressed by interviewees that affect all LGBTI groups are related to an increase in psychosocial support and mental health care regarding LGBTI issues, contact points specialized on the specific needs of each LGBTI group as well as permanent contact people for LGBTI concerns in medical centers. Additionally, an LGBTI certificate for health care professionals trained in LGBTI issues was recommended.

These results show that LGBTI people receive insufficient attention in health care and wish to be protected from discriminatory practices. Previous research shows that especially subtle forms of discrimination are still common, e.g., via concealment or heteronormative assumptions [9,17,18,50]. The research project HEALTH4LGBTI by the European Union, which was also based on expert interviews and focus groups, revealed that access to appropriate medical services is often impeded for LGBTI people, and that there is a need for visible and identifiable LGBTI-friendly health care services [17].

Concerning prevention measures, the present study shows that there is a need for adequate cancer, STI/HIV, addiction, suicide and depression prevention for all LGBTI groups. This is in line with previous studies indicating that the smoking prevalence among LGBT people is significantly higher than in the general population, with bisexual and transgender people being at the highest risk for tobacco use [51]. Alcohol and drug abuse are also increased in LGBT populations [14,26,52,53,54]. According to the minority stress model, the higher smoking prevalence results from minority stress caused by internalized homonegativity and victimization that increase psychological distress [28]. LGBTI-tailored tobacco prevention programs, such as cessation classes, are also cancer prevention programs [51].

Apart from substance abuse, minority stress is also associated with a higher risk of mental health problems among LGBT populations, such as depression and suicide ideation [55,56,57,58,59,60,61]. Suicide prevention programs should support LGBTI people in developing a sense of belonging and improving self-esteem, e.g., via affirmative approaches and trainings for health care professionals to raise awareness on LGBTI issues [57]. There is evidence that LGBTI people are at increased risk for certain cancers and that screening programs do not sufficiently reach this population [14,17,62,63,64,65]. However, there is still a lack of much needed LGBTI cancer research and programs [62]. With regard to STI transmission, current research shows that gay men are at higher risk of HIV than other groups of people, that rates of HIV and other STIs are higher among transgender people than non-transgender people, and that there is a lack of research on the transmission of HIV among lesbian women [65,66,67,68]. This underlines the need for more and extended prevention programs indicated by the present study, such as “The Last Drag”, the first known smoking cessation program designed for LGBT smokers [51] and other smoking cessation programs for LGBTI people that have proved to be successful [52], or “Start Talking. Stop HIV”, a campaign for gay and bisexual men that aims to increase HIV-related communication and knowledge [69].

In the context of diagnostic investigation, treatment and counselling, the present study reveals a wish of LGBTI people for authenticity and openness, e.g., concerning a lack of knowledge, issues of coming-out, and mental stress, and a wish for the depathologization and elimination of ignorance of LGBTI concerns. This goes in line with the results of previous studies indicating a need for affirmative approaches and the training of health professionals [9,17,50,70,71]. Some interventions with LGBT or LGBTI content have already been researched and proven to be successful [72,73,74].

### 4.2. Needs of LGB People

With regard to health care structures, the present study indicates that there is a need for more information materials on lesbian health issues, more research in the field of lesbian health, more visibility of health care services that explicitly include bisexual people as a target group, and low-level counselling services for bisexual people, especially those who struggle with insecurities after becoming aware of their attraction to both male and female individuals. This contributes to previous research indicating that lesbians have more health risks but use preventive medical care less often and receive less quality care than other women [75,76]. It reflects research indicating that bisexual people represent an often ignored subgroup among gender and sexual minorities [77], that they experience even more health inequalities and minority stress due to biphobia in both heterosexual and gay and lesbian communities [17], and that bisexual women are less likely to disclose their identity than lesbian women [78]. Thus, bisexual people, on the one hand, rarely make themselves visible in health care and, on the other hand, no visible bisexual friendly health care services are offered. This indicates a great need for health care professionals with persistent awareness, an open attitude, and specific knowledge, which are factors that are associated with quality care for lesbian clients [79]. We also know from previous research that disclosure is associated with better outcomes and an improved quality of care [78].

As for prevention measures, the results of this study show that there is a risk of lesbian women to be disregarded concerning cancer screening because they do not go to gynecologists as often as heterosexual women. This is consistent with previous research that points to the increased risk of lesbian and bisexual women of developing cervical cancer compared to other women. The issue of gay and bisexual men being at much higher risk of anal cancer than the general male population was not mentioned in the present study but has to be considered in the development of screening programs, too [17]. The results of the present study also indicate that drug prevention for gay men should address the risks of sexualized drug use (chemsex). According to a study by Pufall et al. [80] chemsex is associated with “self-reported depression/anxiety, smoking, nonsexual drug use, risky sexual behaviours, STIs, and hepatitis C”.

In view of treatment procedures, the results of this study point to the need for the elimination of discrimination, so that, for example, lesbian women get the same access to fertility treatment and the same trust in their abilities of being good mothers as heterosexual women, and that gay men are not treated differently from heterosexual men, e.g., at the proctologist. Moreover, this study indicates that health professionals who treat gay men must not be ashamed of talking about anal intercourse and that health professionals who treat bisexual people fully need to accept bisexuality as a distinct sexual orientation.

### 4.3. Needs of Transgender and Intersex People

Concerning health care structures, the present study points to a lack of low-level health care services, couple therapy and gynecological services for transgender people. Moreover, the results show that there is a need for more physicians and psychotherapists who treat transgender people because current waiting times are too long. Both transgender and intersex people express a need for the quality management of health care services and payment and settlement systems of health insurance that do not directly link certain services to a specific gender, e.g., testosterone substitution only to women. Especially for intersex people, there is a need for increasing peer support networks and giving patients access to their complete patient records. This study also reveals a demand for services that offer care for neovaginas. This contributes to research indicating that transgender individuals experience procedures that are necessary for medical transition as arduous, challenging and very complex [8,23,81,82,83]. It also contributes to findings that reveal barriers for LGBTI people when accessing health care, e.g., due to a lack of specialist mental health services and counselling services, unrecognized needs, a lack of relevant documents and protocols, and the use of pathologizing language and incorrect pronouns [17,18]. Moreover, it is in line with research that indicates a massive lack of counselling services for intersex people and parents of children with intersex conditions and a great wish especially for more peer support [84].

Regarding prevention measures, the results indicate a need for gynecological and urological prevention measures that address transgender and intersex people adequately. Furthermore, this study shows that for transgender people, the prevention of harm is important, e.g., in connection with the use of breast binders. The present study also shows that for intersex people, down-to-earth counselling regarding degeneration risk is relevant, neither exaggerating nor downplaying the risks. This underlines and expands previous research that points to the specific but insufficiently researched cancer risks for transgender people, e.g., due to hormone treatment [85,86], and intersex people, e.g., due to early fetal germ cells [87,88].

In the context of diagnostic investigation and treatment procedures, this study shows that it has to be taken into consideration that the reference values for males and females might not be valid for intersex and transgender people. Moreover, the results indicate a need for careful differential diagnostics, careful management of comorbidities, and careful assessment of interdependencies between somatic health aspects and psychological aspects. Regarding transgender people, this study points to the need for an unbiased treatment indication, the empowerment of clients, adequate support during treatment, and treatment procedures that are individually tailored to the needs of each client, including health care services that ensure the integration of physical changes into the whole self. These findings extend previous studies that indicate that transgender clients are very interested in support according to individual needs and high decision-making power concerning the treatment process [23], and that point to the higher prevalence of mental health problems, especially affective and anxiety disorders, among transgender people than the general population [53,68,89,90]. With regard to intersex people, this study points to a need for preventing abusive curiosity in the context of a rare disease, stopping the harmful treatment practices of the past, informing about all the treatment options, non-treatment included, and providing continuous support and practical psychosocial counselling. This underlines previous research that reveals the need for more knowledge and sensitivity on the part of medical professionals and the need for local medical and psychosocial support structures that are readily accessible [84,91]. 

### 4.4. Limitations

The findings of the present study might be limited because the pre-understanding of the researchers might have biased the questions that were posed in the interviews and focus groups. In order to counteract such biases, not only scientific studies but also grey and community literature was included in the literature research. In this way, blind spots were supposed to be detected. Since all qualitative studies can only have a limited number of participants, the generalizability of this study’s results can be questioned. It is not possible to determine unambiguously whether all typical aspects were mentioned that have an impact on how LGBTI people feel and think regarding health services. However, professional literature and previous research indicate that the challenges and problems that were the subject of discussion are characteristic. Moreover, choosing experts who had a function as key people was supposed to ensure a certain generalizability, as they were familiar with a wide range of counselling experiences and could talk about issues that went far beyond their personal perspective. But certainly, the whole range of concerns of LGBTI people is not represented in this study. For example, women with transition experiences (transgender women) and people who do not have any connections to the LGBTI scene did not take part. Furthermore, certain topics were almost left out, e.g., care for older LGBTI people, questions of intersectionality (e.g., the situation of LGBTI people from a migration background or with disabilities) or the black market of phosphodiesterase type 5 (PDE-5) inhibitors. Specific health concerns of, for instance, pansexual, asexual or questioning people were not considered either. In addition, this study was limited to the Hamburg metropolis, due to the contracting authority. Thus, the experiences of LGBTI people who live in rural areas are not represented. Further research is needed in this respect.

Finally, qualitative content analysis according to Mayring [49] is a rather schematic analysis method so that the underlying pre-understanding of the researchers is not without relevance. This limitation was supposed to be minimized by building the categories inductively.

## 5. Conclusions

The present study demonstrates that, essentially, it is important to recognize LGBTI individuals and their needs. This is about recognizing depathologization and dealing consciously, sensitively and inclusively with LGBTI people in the health system. This can be done by, for example, visibly marking LGBTI expertise (e.g., with a rainbow sticker on the door label), or by actively including LGBTI individuals in cancer prevention programs (e.g., through mandatory invitations to cancer screening for all). Communication training (e.g., for gender-sensitive and integrative use of language), in-service training (e.g., for nursing staff) and sufficient opportunities for LGBTI-competent counselling and psychosocial care are equally important. General health care professionals should have some basic knowledge about LGBTI health issues, such as STI transmission, risks for certain mental problems, and somatic aspects of transgender and intersex clients. Beyond that, more psychotherapeutic services based on affirmative approaches are needed for LGBTI clients. Moreover, there is a lack of health care services with specific expertise for transgender and intersex clients.

## Figures and Tables

**Table 1 ijerph-16-03547-t001:** Overview of main categories and subcategories.

Experts and Focus Groups	EL	EG	EB	ET	EI	Focus Group 1:	Focus Group 2:	Focus Group 3:	
**Main and Subcategories**						**Approach**	**Attitude**	**Community**	**In Total**
1. Requests—people	1	0	0	4	1	6	12	2	26
2. Requests—structures	1	5	6	0	2	10	6	3	33
3. Training (incl. further education)	2	4	0	0	2	11	10	4	33
4. Public image	2	4	1	1	1	10	1	11	31
5. Treatment procedures	3	1	2	4	3	12	12	3	40
6. Diagnostic procedures	0	3	1	7	2	7	0	0	20
7. Discrimination	6	4	0	1	6	11	7	6	41
8. Living worlds/situations/realities									
8.1 Acceptance	0	4	2	0	0	3	3	5	17
8.2 Coming-out	0	3	1	0	0	2	2	1	9
8.3 Diversity of life forms	0	3	3	1	0	1	0	2	10
8.4 Life stages	0	2	1	0	0	2	0	1	6
8.5 Sex	1	6	1	0	1	0	0	0	9
8.6 Scene/Community	0	2	3	0	0	2	0	11	18
9. Networks									
9.1 Address lists	1	1	1	0	1	0	0	2	6
9.2 Interlocking	0	4	0	0	0	0	0	1	5
9.3 Referral	0	1	0	2	0	0	0	1	4
9.4 Mouth-to-mouth	0	0	0	0	0	3	0	4	7
10. Openness/Willingness to communicate	4	4	0	1	1	7	12	23	52
11.Economics	0	3	0	0	1	1	4	0	9
12. Prevention measures	1	2	0	3	1	0	0	1	8
13. Sensitization (Attitude)	3	0	3	5	1	16	24	19	71
14. Sexually transmitted infections	4	6	0	0	0	4	0	4	18
15. (In-)Visibility	2	0	0	1	1	6	4	6	20
16. Availability (lacking)	3	1	4	5	4	3	7	6	33
17. Bias/prejudices	1	1	2	1	0	1	0	2	8
In total	35	64	31	36	28	118	104	118	534

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
