# Peer review of "The Needs of LGBTI People Regarding Health Care Structures, Prevention Measures and Diagnostic and Treatment Procedures: A Qualitative Study in a German Metropolis"

_ijerph, 2019, doi:10.3390/ijerph16193547_

Round 1

Reviewer 1 Report

The study is interesting but it could be important to systemise the qualitative part in a syntetic part

Author Response

Response to Reviewer 1

Thank you very much for reviewing the manuscript and for your feedback concerning the qualitative part.

...it could be important to systemise the qualitative part in a syntetic part

In the methods section we now added some more details concerning the steps of analysis. In the results section we added some more quotations in order to convey a more complete image of this qualitative research. We hope that this has also contributed to make it more systematic and syntetic in the sense of your feedback.

New/extended text passages:

According to Denzin’s [48] basic types of triangulation, the present study ensured data triangulation via multiple data sources, i. e. experts who were interviewed because of their function as multiplier key persons and persons in their role as private individuals who took part in focus groups. Investigator triangulation was accomplished via integrating three researchers in the process of data collection, data analysis and interpretation. Methodological triangulation was assured when using two different qualitative methods, i. e. expert interviews and focus groups.[...]

All of the interviews were conducted by the same researcher of the team (UL) to ensure a comparable standard. A second researcher of the team (PiB) took part as a participant observer to reveal potential differences in the approach of the interviewees and decrease potential biases in the further research process.[...]

The interviews were read several times by three researchers (UL, PiB, TN). Based on the research questions, these three researchers built a system of categories by initial coding and rechecking the interview material. In the end, the results were put together thematically in order to frame the main and subcategories. Frequent comparisons and adjustments of the results by the three independent assessors ensured the interrater reliability of the results.[...]

The student assistant (PiB) documented the process as a participating observer in order to ensure that possible biases due to the personalities of the moderators were revealed and decreased in the further process of investigation.

New/extended text passages:

In 2017 the senate of Hamburg adopted an action plan for gender and sexual diversity which contained eleven areas of activity for people between childhood and ninety years of age. The plan was a token of tolerance and openness in Hamburg. Financial resources were provided for 90 measures. With regards to content, the plan’s aims were information, education, sensitization, making different concepts of living more common, and protection of rights [43]. The present study was part of this action plan.[…]

EG:“There is a severe shortage of Psychologists and psychiatrists in Hamburg [especially compared to Hamburg’s well established network of specialized medical practices for HIV/STI, UL].”[…]

F1G:“[...] the largest operator of seniors' and nursing homes in Munich just started a model project [...] and presents the rainbow flag on its website [...]. Unfortunately, this does not mean much, as it is still unknown how gays, lesbians, inter- and transpeople are approached. It is different in Holland. They have a certification procedure [...], which is also examined by a third party.”[…]

She stresses that there is a lack of respective services especially in Hamburg:

EL:“Well, in terms of fertility treatment, Hamburg lags behind Munich and Erlangen - and Berlin anyway.”

EL:“And there are countries where I assume that they are not lesbian-friendly, but where reproductive medicine is still handled openly and liberally, so there are no barriers.”[…]

All in all, the participants did not differ very much from each other in their positions.

Reviewer 2 Report

The authors make it clear that the results of this study are specific to Hamburg, Germany. However, there is no real attempt to situation the results from the expert interviews and focus goups in relationship to the specific social and economic conditions in Hamburg. Is it possible to provide doctors and medical personnel that are sensitive to LGBTI concerns? The results were interesting, but seem to be pretty consistent across western countries. I also wondered about issues such as race and social class with both the expert witnesses and the focus groups. The people interviewed seemed very articulate about the problems they faced and felt entitled to these problems being addressed. 

Author Response

Response to Reviewer 2

Thank you very much for reviewing our manuscript and for your precious feedback!

However, there is no real attempt to situation the results from the expert interviews and focus goups in relationship to the specific social and economic conditions in Hamburg. Is it possible to provide doctors and medical personnel that are sensitive to LGBTI concerns?

We now added some information on the specific situation in Hamburg in the introduction and in some other passages of the text.

New/extended text passages:

In 2017 the senate of Hamburg adopted an action plan for gender and sexual diversity which contained eleven areas of activity for people between childhood and ninety years of age. The plan was a token of tolerance and openness in Hamburg. Financial resources were provided for 90 measures. With regards to content, the plan’s aims were information, education, sensitization, making different concepts of living more common, and protection of rights [43]. The present study was part of this action plan.[…]

EG:“There is a severe shortage of Psychologists and psychiatrists in Hamburg [especially compared to Hamburg’s well established network of specialized medical practices for HIV/STI, UL].”[…]

She stresses that services are lacking especially in Hamburg:

EL:“Well, in terms of fertility treatment, Hamburg lags behind Munich and Erlangen - and Berlin anyway.”

EL:“And there are countries where I assume that they are not lesbian-friendly, but where reproductive medicine is still handled openly and liberally, so there are no barriers.”

I also wondered about issues such as race and social class with both the expert witnesses and the focus groups.

We extended the limitations sections and mentioned that questions of intersectionality were not enough represented in our study.

Changed text: Furthermore, certain topics were almost left out, e.g. care for older LGBTI people, questions of intersectionality (e.g. the situation of LGBTI persons from a migration background or with disabilities) or the black market of PDE-5 inhibitors.

Reviewer 3 Report

Thank you for sharing this paper for review. I enjoyed the read and I think that the paper could be a very good contribution on LGBTI health. However, my principal concern is related with the qualitative design and how it is explained. Following you can find some questions and advices that you can follow to improve your paper.

Although the research problem is informed by a review of the relevant empirical work, the critical issues is that the paper is not clear about the underpinning theoretical stance beyond the testimonial approach of the current study. Author/s should ensure that there is an explicit alignment and consistency between a theoretical stance and their approach considering what factors can “influence” or be related with LGBTI-related health and how we can learn about them through the testimonial approach. They should made it explicit in the introduction of the paper. I have missed several references about qualitative research on LGBTI health that has been conducted during the last years from a qualitative perspective. A brief review of the qualitative research could improve the understanding of how the current study adds to the LGBTI health research. . What criteria was used to select the five experts participating in the interviews? Questions presented in the interviews and focus subsections are the total number of questions include in the interviews or the interview script included additional questions? It will be useful to include the final interview script annexed in order to allow replicability in order cultural contexts. I would like to see how triangulation was conducted in the selection of the narratives. I am probably wrong but after reading the selection procedure I had the impression that the selected narratives were chosen because they include similar thoughts in relation to the questions asked. Indeed, I found that in the results analysis there is a lack of criticality. A convincing analysis should explicitly seeks out and test rival explanations and counter examples. Data analysis require a more detailed description to allow readers to follow each step of the analysis. For example, did the author/s used triangulation of data in the categorization procedure? Please, explain how triangulation was assured in each phase described. It should be explain that data triangulation means using data from different participants or in different settings or at different times. Please, clarify. The conclusion section is lack of previous references. I would like to see how the analyses of the testimonials selected could help in term of practical implications improving LGBTI-related health issues.

Author Response

Response to Reviewer 3

Thank you very much for all your comments and your valuable hints to improve our paper! We tried to take all of these proposals into account.

Although the research problem is informed by a review of the relevant empirical work, the critical issues is that the paper is not clear about the underpinning theoretical stance beyond the testimonial approach of the current study. Author/s should ensure that there is an explicit alignment and consistency between a theoretical stance and their approach considering what factors can “influence” or be related with LGBTI-related health and how we can learn about them through the testimonial approach.

We added a text passage on the approach of the current study.

New text: The present study took a qualitative approach. We were interested in specific and detailed experiences of professionals and/or clients in the healthcare system, and aimed at discovering and understanding how the participants view their living conditions under these circumstances. Thus a person-centered and participatory approach seemed to be appropriate to learn more about factors influencing LGBTI health in Hamburg [46,47]. Main purpose was to collect what problems and challenges LGBTI face in Hamburg’s health care.

I have missed several references about qualitative research on LGBTI health that has been conducted during the last years from a qualitative perspective. A brief review of the qualitative research could improve the understanding of how the current study adds to the LGBTI health research.

We added further references and pointed out the additional value of the present study.

Changed text: Several international and national qualitative studies have already investigated healthcare needs of LGBTI people [e.g. 9,17–22]. However, the present study is the first to investigate the healthcare needs of LGBTI people in Hamburg, a major German metropolitan city. Thus it is still unknown if the results of existing studies also prove true for healthcare in Hamburg.

New text: Compared to previous qualitative research on the needs of LGBTI persons the results of the present study are quite comparable. Fear of discrimination, lack of knowledge of health care professionals, higher risk of mental health problems, fear of disclosure, being confronted with unquestioned heteronormative assumptions, importance of visibility, and pathologization were also highlighted as important issues in previous studies conducted in the European Union [17,18], Australia [20], and the United States [19]. By contrast with research from Zimbabwe problems of stigmatization, discrimination, ill-informed personnel and lacking access to healthcare were much less prominent and serious, although also reported in Hamburg [21,22]. By the present study previous evidence is extended and enriched by further qualitative content, e.g. personal experiences and concrete examples, from Hamburg, Germany.

Further references:

Alpert, A.B; CichoskiKelly, E.M.; Fox, A.D. What lesbians, gay, bisexual, transgender, queer, and intersex patients say doctors should know and do: A qualitative study. J Homosex 2017; 64, 1368–89.

Ansara, Y.G. Challenging cisgenderism in the ageing and aged care sector: Meeting the needs of older people of trans and/or non-binary experience. Australas J Ageing2015, 34, 14–18.

Hunt, J.; Bristowe, K.; Chidyamatare, S.; Harding, R. ‘They will be afraid of you’: LGBTI people and sex workers’ experience of accessing healthcare in Zimbabwe – an in-depth qualitative study. BMJ Glob Health2017, 2, e000168.

Hunt, J.; Bristowe, K.; Chidyamatare, S.; Harding, R. ‘So isolation comes in, discrimination and you find many people dying quietly without any family support’: Accessing palliative care for key populations – an in-depth qualitative study. Palliat Med2019, 33, 685–692.

What criteria was used to select the five experts participating in the interviews?

This is explained in the methods section.

Text passages:

For the expert interviews, professional key persons of the focused target groups were recruited in the Hamburg area, who next to their own healthcare experiences knew of experiences of other people in the respective group with medical and mental health professionals (as a multiplier).[…]

The focus groups were conducted in the same way as the expert interviews (see above); representatives of the LGBTI groups and experts from professional practice were contacted. Since the focus was on personal experiences, the function as a multiplier key person was not a prerequisite for participation.

Questions presented in the interviews and focus subsections are the total number of questions include in the interviews or the interview script included additional questions? It will be useful to include the final interview script annexed in order to allow replicability in order cultural contexts.

The interview guideline is translated into English and can be found in the appendix.

New text:

Appendix A

Interview Guideline: „What does diversity have in common? On equality between women, men and LGBTI people using health care in Hamburg“

Research question: What are the challenges and problems of health promotion and health care for women, men and LGBTI people in Hamburg?

Lead text

Information on the goals of the study:Questioning the extent to which equality in health promotion and health care is ensured in Hamburg and at which points shortcomings exist which need to be addressed. In particular, it should be examined whether gender and/or sexual orientation have an impact on health care.

Information on the course of the study:Orientation on the guideline, which should ensure that the interviews are comparable. Reference to colleague Pia Behrendt, who pays attention to the comparability and who documents relevant differences. Enquiries are possible so that no topic is omitted. Altogether it is about exemplary experiences, which can reflect personal and typical experiences.

Audio Recording Information:Consent to transcription and pseudonymization.

Data Protection Information:No recording of personal data that would enable inferences about specific interviewees.

Now I would like to start by asking you a few short questions about your role as an expert, as which I am interviewing you today…

TOPIC: Expert function of the interviewee

Research question: What makes the interviewee an expert?

Concrete interview questions

We have chosen you against the background of your activities in the context of ____________________  as expert for health promotion and health care for women/men/lesbian women/gay men/transgender persons/intersexuals. Are there any other functions that qualify you as an expert for this topic?

Since when do you execute this function(s)?

Are there one or more main age groups that you mainly deal with in this function (these functions)? If so, which?

In the following, I will first ask questions about the experience of concrete treatment or counselling situations, then I’ll continue with questions about the specific knowledge of the treatment or counselling personnel and finally follow two questions about existing information needs... 

INTERVIEW PART I: OBSERVABLE DISADVANTAGES IN CONCRETE TREATMENT/ADVICE SITUATIONS?

TOPIC: Relationship management

Research question: How do the different groups of people experience relationship management on the part of the treatment or counseling personnel?

Check / Memos

Concrete interview questions

Maintenance questions

- sensitive language

- recognition of individuality

- sufficient self-awareness/reflection on the part of the practitioners/consultants

- good ability to talk openly about sexual orientation/sexuality

- disclosure of one's own way of life?

- promotion of resilience, resources, self-confidence

- caring behaviour vs. promotion of self-determination

What do you know about how women/men/LGBTI people experience the relationship with the practitioners/consultants?

What do you know about how women/men/LGBTI people experience the behaviour of practitioners or consultants towards them?

- concrete (positive and negative) experiences?

- concrete suggestions for improvement?

What do you know about, how women/men/LGBTI people feel in treatment or counselling situations?

What other experiences can you report?

What other experiences can you describe?

TOPIC: Practices and structures

Research question: To what extent do health care practices and structures provide the best possible treatment/counselling for all groups of people?

Check / Memos

Concrete interview questions

Maintenance questions

- Knowledge of competent contact persons (referral)

- Legal knowledge

- formal discrimination*

Do you assume that women/men/LGBTI persons always receive the best possible treatment or counselling? To what do you attribute this?

- concrete (positive and negative) experiences?

- concrete suggestions for improvement?

What do you know about when women/men/LGBTI persons have been (very) satisfied with treatment or counselling?

Or (very) dissatisfied, and why was that?

* includes disadvantage or exclusion in treatment processes or lack of access to rights and resources (vs. informal discrimination that affects verbal or non-verbal conduct that offends, excludes and impairs the integrity and well-being of the individuals)

TOPIC: Attitude towards the group of persons

Research question: What is the attitude towards the different groups of people?

Check / Memos

Concrete interview questions

Maintenance questions

- affirmative attitude

- inclusive thinking

- prejudices

- LGBTI-friendly environment

- general openness for diversity

- interpersonal discrimination*

What do you know about the attitudes of treatment and counselling staff towards women/men/LGBTI persons?

- concrete (positive and negative) experiences?

- concrete suggestions for improvement?

What (particularly) positive or (particularly) negative experiences are you aware of that have been made by women/men/LGBTI persons, and what exactly has happened?

*refers, for example, to the mood that is transported verbally, the number of eye contacts, the time taken by the practitioners/consultants

INTERVIEW PART II: LACK OF KNOWLEDGE ABOUT EACH GROUP OF PERSONS

TOPIC: Expertise on the specific health topics of the respective groups of people

Research question: To what extent do health care professionals have sufficient expertise on the specific health issues of the groups?

Check / Memos

Concrete interview questions

Maintenance questions

- somatic health issues

- mental health issues

- knowledge about sexual orientation/sexuality

- relevant transmission pathways of HIV/STI

- cancer screening

- importance of smoking, alcohol, drugs

Do you have the impression that the treatment or counselling staff is sufficiently aware of the specific health concerns of women/men/LGBTI people? To what do you attribute this?

- concrete (positive and negative) experiences?

- concrete suggestions for improvement?

When did you discover or learn that specific knowledge was helpful or necessary - or would have been?

Imagine someone has a health question that concerns the person as a woman/man/LGBTI person. Who would you recommend as a contact person?

TOPIC: Assumptions on aetiology

Research question: What assumptions, which are specifically related to gender and/or sexual orientation, does the treatment or counselling staff have regarding the aetiology of diseases/disorders?

Check / Memos

Concrete interview questions

Maintenance questions

- (de-)pathologization

- aetiology of the development of sexual orientation/sex

- attitude towards conversion or reparative therapies

When do you have the impression that the treating or advising person is explaining a so-called disease or disorder with the help of sex or sexual orientation?

- concrete (positive and negative) experiences?

- concrete suggestions for improvement?

What other examples can you think of where gender or sexual orientation is used to explain a disease or disorder?

TOPIC: Life reality of the individual groups of people

Research question: To what extent is there an awareness in health care of the reality of life of the various groups of people?

Check / Memos

Concrete interview questions

Maintenance questions

- experiencing otherness

- coming out process

- community/scene

- parenthood

To what extent do you have the impression that the practitioner or counsellor knows enough about the life situation as a woman/man/LGBTI person?

- concrete (positive and negative) experiences?

- concrete suggestions for improvement?

Do you think that the life reality of women/men/LGBTI persons is sufficiently taken into account? Why?

INTERVIEW PART III: LACK OF AVAILABILITY OF SPECIFIC INFORMATION FOR THE RELEVANT GROUP OF PERSONS

TOPIC: Provision of target group-specific information

Research question: To what extent is specific information made available for the respective groups of people in the health care system?

Check / Memos

Concrete interview questions

Maintenance questions

- information on target-group-specific treatment/consulting offers

- flyers/brochures and the like

- access to treatment/counselling

- visibility of the groups of persons

In your opinion, how well is information provided about health services that specifically concern you, women/men/LGBTI?

- concrete (positive and negative) experiences?

- concrete suggestions for improvement?

Where do you get information about special health offers for you as a woman/man/LGBTI person?

Imagine you have a health question that specifically concerns someone as a woman/man/LGBTI person. Where would you look for information?

What information do you know about special health offers for you as a woman/man/LGBTI person?

TOPIC: Raising awareness of groups of people for their own health issues

Research question: To what extent are there efforts in health care to sensitise the various groups of people to their health issues?

Check / Memos

Concrete interview questions à LAST QUESTION

Maintenance questions

- availability of information in online sources

- availability of information in brochures

- information provided by the practitioner/counsellor

How well do you think women/men/ LGBTI persons feel about health-relevant topics (e.g. prevention of diseases) that specifically concern you as a woman/man/LGBTI person?

- concrete (positive and negative) experiences?

- concrete suggestions for improvement?

Where do women/men/LGBTI-persons get information on health-relevant topics (e.g. prevention of diseases), which especially concern you as a woman/man/LGBTI-person?

What kind of health-relevant information (e.g. on the prevention of illnesses) do you know that is especially targeted to women/men/LGBTI people?

CONCLUSION

Are there still important aspects of the topic that have not been considered enough in the previous interview?

Would you like to add anything else?

_______________________________________________________________

I would like to see how triangulation was conducted in the selection of the narratives. [...] Data analysis require a more detailed description to allow readers to follow each step of the analysis. For example, did the author/s used triangulation of data in the categorization procedure? Please, explain how triangulation was assured in each phase described. It should be explain that data triangulation means using data from different participants or in different settings or at different times. Please, clarify.

In the first paragraph of the section "2. Materials and Methods" and in the last paragraph of the section "2.1 Expert interviews" and the last paragraph of the section "2.2 Focus groups" we now describe the procedure of analysis in greater detail.

New/extended text passages:

According to Denzin’s [48] basic types of triangulation, the present study ensured data triangulation via multiple data sources, i. e. experts who were interviewed because of their function as multiplier key persons and persons in their role as private individuals who took part in focus groups. Investigator triangulation was accomplished via integrating three researchers in the process of data collection, data analysis and interpretation. Methodological triangulation was assured when using two different qualitative methods, i. e. expert interviews and focus groups.[...]

All of the interviews were conducted by the same researcher of the team (UL) to ensure a comparable standard. A second researcher of the team (PiB) took part as a participant observer to reveal potential differences in the approach of the interviewees and decrease potential biases in the further research process.[...]

The interviews were read several times by three researchers (UL, PiB, TN). Based on the research questions, these three researchers built a system of categories by initial coding and rechecking the interview material. In the end, the results were put together thematically in order to frame the main and subcategories. Frequent comparisons and adjustments of the results by the three independent assessors ensured the interrater reliability of the results.[...]

The student assistant (PiB) documented the process as a participating observer in order to ensure that possible biases due to the personalities of the moderators were revealed and decreased in the further process of investigation.

I found that in the results analysis there is a lack of criticality. A convincing analysis should explicitly seeks out and test rival explanations and counter examples. 

We added further quotations in order to convey a more complete picture of this qualitative research. The interviewees do not differ very much from each other in their opinions. The difference mainly lies in the different examples the interviewees mention. The aim of the study mainly was to gather all of these problems and challenges in LGBTI health care in Hamburg. The interview questions were adapted to this aim. It was not one of our aims to point out different views in each one of the five groups of the LGBTI spectrum but to look for similarities and differences between these five groups.

New text passages:

F1G:“[...] the largest operator of seniors' and nursing homes in Munich just started a model project [...] and presents the rainbow flag on its website [...]. Unfortunately, this does not mean much, as it is still unknown how gays, lesbians, inter- and transpeople are approached. It is different in Holland. They have a certification procedure [...], which is also examined by a third party.”[…]

She stresses that there is a lack of respective services especially in Hamburg:

EL:“Well, in terms of fertility treatment, Hamburg lags behind Munich and Erlangen - and Berlin anyway.”

EL:“And there are countries where I assume that they are not lesbian-friendly, but where reproductive medicine is still handled openly and liberally, so there are no barriers.”[…]

All in all, the participants did not differ very much from each other in their positions. The difference depended mainly on the examples given by the interviewees.

I would like to see how the analyses of the testimonials selected could help in term of practical implications improving LGBTI-related health issues. 

In the conclusion section we added some more specific practical implications.

Changed text: This is about recognising depathologization and dealing consciously, sensitively and inclusively with LGBTI persons in the health system. This can be done by, for example, visibly marking LGBTI expertise (e.g. with a rainbow sticker on the door label), or by actively including LGBTI individuals in cancer prevention programmes (e.g. through mandatory invitations to cancer screening for all). Communication training (e.g. for gender-sensitive and integrative use of language), in-service training (e.g. for nursing staff) and sufficient opportunities for LGBTI competent counselling and psychosocial care are equally important.

Round 2

Reviewer 2 Report

I feel that the authors have addressed my concerns that the original submission was too general in nature and didn't justify the focus on the one city. 

Author Response

Thank you very much again for your feedback and your helpful recommendations for improvement!

Reviewer 3 Report

Authors have addressed all my concerns in a satisfactory way. The qualitative desing and methodology is now well described allowing replicability. Triangulation has now been assured. Results are well written and the conclusions are built on the results obtained. In my opinion the article merits publication.

Author Response

(The authors gave the same response as above.)
